# Citrus Identification and Counting Algorithm Based on Improved YOLOv5s and DeepSort

**Yuhan Lin** **, Wenxin Hu, Zhenhui Zheng and Juntao Xiong \***

School of Mathematics and Information Science, South China Agricultural University, Guangzhou 510642, China;
lyh_hans@163.com (Y.L.); q1158653026@163.com (W.H.); zzh083@stu.scau.edu.cn (Z.Z.)
\* Correspondence: xiongjt@scau.edu.cn; Tel.: +86-135-6016-4695

**Abstract:** A method for counting the number of citrus fruits based on the improved YOLOv5s algorithm combined with the DeepSort tracking algorithm is proposed to address the problem of the low accuracy of counting citrus fruits due to shading and lighting factors in videos taken in orchards. In order to improve the recognition of citrus fruits, the attention module CBAM is fused with the backbone part of the YOLOv5s network, and the Contextual Transformer self-attention module is incorporated into the backbone network; meanwhile, SIoU is used as the new loss function instead of GIoU to further improve the accuracy of detection and to better keep the model in real time. Then, it is combined with the DeepSort algorithm to realize the counting of citrus fruits. The experimental results demonstrated that the average recognition accuracy of the improved YOLOv5s algorithm for citrus fruits improved by 3.51% compared with the original algorithm, and the average multi-target tracking accuracy for citrus fruits combined with the DeepSort algorithm was 90.83%, indicating that the improved algorithm has a higher recognition accuracy and counting precision in a complex environment, and has a better real-time performance, which can effectively achieve the real-time detection and tracking counting of citrus fruits. However, the improved algorithm has a reduced real-time performance and has difficulty in distinguishing whether or not the fruit is ripe.

**Keywords:** YOLOv5s; DeepSort; attention mechanism; citrus detection; loss function; CoTNet

## 1. Introduction

In the context of the national support for rural revitalization and the rapid development of smart agriculture, the yield estimation of on-tree fruits during the growing period, thus enabling the smart management of orchard production, has become a research hotspot. China is one of the major citrus-growing countries in the world, and in the process of citrus picking and conducting citrus fruit counts; thus, estimating citrus yields is important for guiding managers in orchard management and formulating citrus planting and marketing strategies.

For the fruit recognition problem, many researchers at home and abroad have proposed many solutions based on traditional machine learning algorithms [1]. Lin G. et al. [2] proposed a novel detection algorithm based on color, depth and shape information, which is less effective in recognizing when occluded by leaves or branches. Chaivivatrakul S. et al. [3] proposed a technique based on a texture analysis to detect green fruits on plants, which is most affected by lighting. Fu L. et al. [4] proposed a red-green-blue-depth (RGB-D) camera using a fruit detection and localization method, which has a higher overall complexity and cost.

The traditional fruit detection identification technology usually has certain limitations in the complex environment adaptability, where it is difficult to meet the actual work needs. The deep learning algorithm has a better adaptability and higher recognition rate, and is gradually becoming the main detection technique today. Numerous researchers have proposed deep learning algorithms such as RCNN [5], FasterRCNN [6], SSD [7], YOLO [8], etc.

In terms of fruit counting, the traditional Sort algorithm has a very obvious problem related to the frequent switching of ID during the counting process, and it is only applicable

to the counting in the case of no occlusion; meanwhile, in the problem of citrus fruit counting, it often happens that the fruit is occluded by other factors such as branches and leaves. Thus, the algorithm is not applicable to the counting of fruit. To address this problem, Wojke N. [9] proposed the DeepSort multi-target tracking algorithm, which was improved based on the traditional Sort algorithm to improve the tracking accuracy of the tracked object in the case of multiple occlusions and to reduce the frequent switching of ID.

To achieve accurate and fast recognition, as well as the accurate counting of citrus in complex environments, this paper improves the YOLOv5s network model by incorporating the attention module CBAM into the model to improve its feature extraction ability. The Contextual Transformer self-attention module is also added to enhance the global extraction ability of the model's deep and shallow features. Moreover, SIoU [10] is used instead of GIoU to introduce the vector angle between the prediction frame and the target frame to redefine the correlation loss function and improve the localization accuracy of citrus fruits. It is also combined with the DeepSort algorithm to realize the counting of citrus fruits. The experimental results demonstrate that the improved algorithm has a significant improvement in both fruit recognition and counting, and can perform the citrus fruit yield estimation more accurately.

## 2. Materials and Methods

### 2.1. Image Acquisition

The images used in this paper were obtained from a citrus orchard at South China Agricultural University, Guangzhou, Guangdong, China (113° E, 23° N), and the image acquisition experiments were conducted on 3 July 2022 and 20 December 2022. As shown in Figure 1, the distance between the camera and the citrus tree trunk is 1–2 m, and the citrus fruit is photographed from multiple angles during the image acquisition process. To meet the diversity of the dataset, images under different lighting conditions were also considered, and were taken from 9:00 a.m. to 20:00 p.m. A total of 1940 citrus images were collected and saved in a JPG format with 640 × 480 resolution.

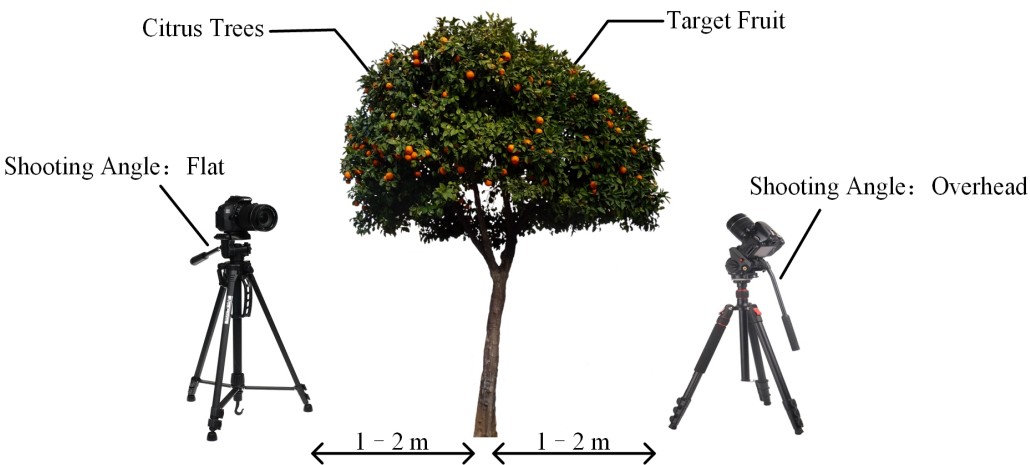

**Figure 1.** Image acquisition schematic diagram.

### 2.2. Building the Dataset

The dataset used in this paper contains 1940 images, which are randomly divided into the training set, validation set and test set according to the ratio of 8:1:1, followed by the manual data annotation and label generation using the LabelImg tool. In addition, when the images in the training set are fed to the detection model, Mosaic data enhancement and the adaptive scaling of the images are automatically performed to enrich the dataset, which effectively improves the generalization capability of the detection algorithm.

## 2.3. General Flow Chart of Fruit Detection and Counting

The flow chart of the proposed method is shown in Figure 2. First, the YOLOv5s network model is improved by adding the CBAM attention mechanism, fusing the CoT3 module in the backbone network and replacing the SIoU loss function in three ways, and a target detection model is trained using the self-built dataset for detecting citrus fruits in images. The video to be detected is fed into the trained citrus fruit detection model as video frames for target detection and followed by feeding the detected target borders and image features into the DeepSort algorithm for fruit counting; finally, the results are displayed as visualizations.

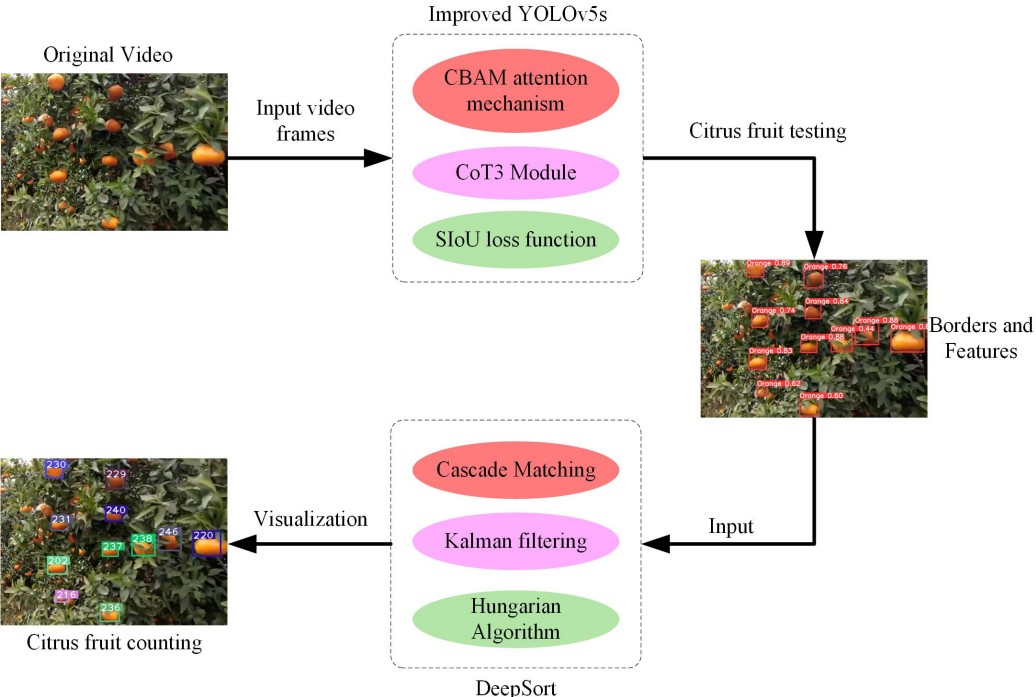

**Figure 2.** Flowchart of the proposed method in this paper.

## 3. YOLOv5 Algorithm

### 3.1. Principle of YOLOv5 Algorithm

The YOLO algorithm is currently the mainstream target detection algorithm, with YOLOv2 [11], YOLOv3 [12], YOLOv4 [13] and YOLOv5 [14], which are a series of algorithms proposed one after another. The YOLO series algorithms meet in real-time at the same time as the detection accuracy improves; the algorithm improvement research also becomes more and more mature. The YOLOv5 series is divided into YOLOv5s, YOLOv5m, YOLOv5l and YOLOv5x [15], depending on the depth and width of the network. In this paper, based on the actual scene requirements and configuration, the YOLOv5s model is used as the base network architecture. It mainly consists of four parts: Input, Backbone, Neck and Output, and its network structure is shown in Figure 3. However, we found that YOLOv5s has some defects during the experiments, such as the missed detection of obscured objects and the wrong detection of background objects. Therefore, this study improves the basic YOLOv5s model based achieving a higher accuracy in citrus fruit recognition, while ensuring the real-time performance of the model.

### 3.2. Improvement of YOLOv5s Algorithm

#### 3.2.1. CBAM Attention Mechanism

In orchards, the shading of branches and leaves as well as the influence of environmental factors such as lighting lead to the low accuracy of the existing YOLOv5 model for the recognition and localization of citrus fruits. To address these problems, the CBAM attention

mechanism is introduced into the YOLOv5 model [16] to improve the model's attention to citrus fruits, ignore unimportant information such as branches and leaves, and make it focus on the current task to improve the efficiency and accuracy of the model in detecting images.

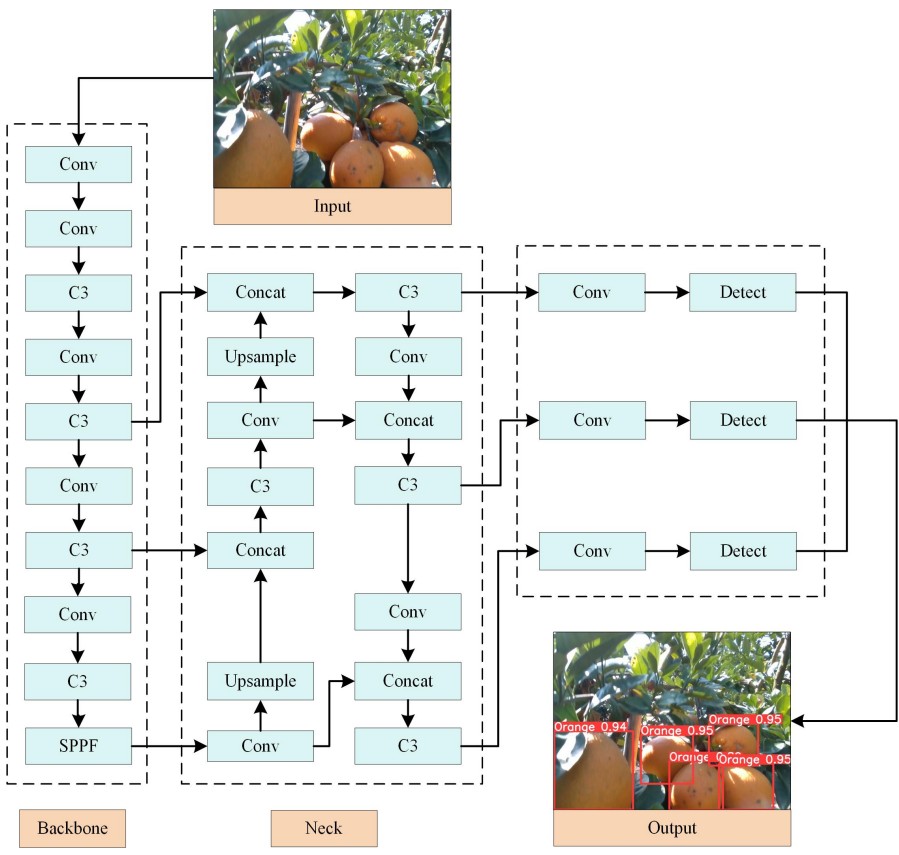

**Figure 3.** YOLOv5s network structure.

The role of the attention mechanism [17] is to allocate the limited information processing resources to the more important parts. Attention mechanisms can be divided into channel attention mechanisms and spatial attention mechanisms. Compared with other attention mechanisms, the advantage of CBAM is that it contains both modules, the channel attention mechanism and spatial attention mechanism [18], and its adaptive feature modification of the feature map through these two sub-modules enhances the importance of the spatial location information and channel information features of the feature map. This improves the detection effect of the network model on the target and improves the representation capability of CNNs, whose structure schematic diagram is shown in Figure 4.

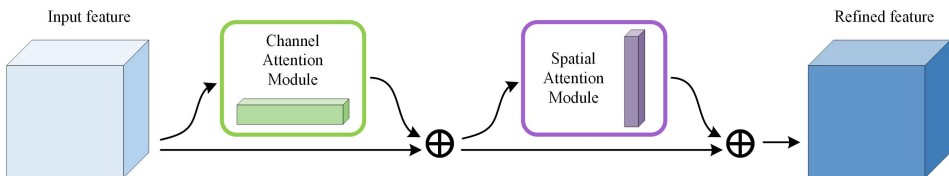

**Figure 4.** Schematic diagram of CBAM structure.

First, the feature map $F(H \times W \times C)$ is input to the channel attention for global max pooling and global avg pooling to obtain two one-dimensional feature maps, which are fed into the neural network (MPL) for computing and then summed to obtain a one-

dimensional channel attention map. This will be multiplied element-wise with F to obtain the channel attention-adjusted feature map $F'$, as shown in Equation (1):

$$F' = M_c(F) \bigotimes F, \tag{1}$$

Then, $F'$ is input to the spatial attention for global max pooling and global avg pooling to obtain two 2D feature maps, which are stitched together and subjected to a convolution operation to obtain a 2D spatial attention map. This is multiplied element-wise with $F'$ to obtain the final feature map $F''$, as shown in Equation (2):

$$F'' = M_s(F) \bigotimes F', \tag{2}$$

In this paper, the CBAM attention mechanism is added before the SPPF layer of the backbone network to enhance the detection of the model in complex backgrounds, so that the model can focus more on the feature extraction of citrus fruits and improve the accuracy of citrus fruit recognition and localization.

3.2.2. SIoU Loss Function

YOLOv5 uses CIoU Loss as the loss function of the Bounding box, while Logits loss function and binary cross entropy (BCE) are used to calculate the loss of the target score and class probability, respectively. The method involves inverse trigonometric functions, which consume some arithmetic power in the process of computation and lead to a slow convergence rate. In addition, the model may "wander around" during the training process, resulting in a worse model and worse detection of citrus fruits.

In this regard, the SIoU loss function is introduced in this paper to measure the deviation between the model output value and the actual observed value in order to help the model fit better, improve the performance of the citrus fruit detection model, and enhance the real-time performance of the algorithm.

SIoU is calculated as shown in Equations (3) and (4):

$$\text{SIoU} = 1 - \text{IoU} + \frac{\Delta + \Omega}{2}, \tag{3}$$

$$\text{IoU} = \frac{|B \cap B^{GT}|}{|B \cup B^{GT}|}, \tag{4}$$

where $B$, $B^{GT}$ denotes the prediction frame and the target frame, $\Delta$ denotes the distance loss recalculated by introducing the angle loss, $\Omega$ denotes the shape loss, and $\Delta$, $\Omega$ is calculated as shown in Equations (5) and (6):

$$\Delta = \sum_{t=x,y} \left(1 - e^{-\gamma \rho_t}\right) = 2 - e^{-\gamma \rho_x} - e^{-\gamma \rho_y}, \tag{5}$$

$$\Omega = \sum_{t=w,h} \left(1 - e^{-w_t}\right)^\theta = \left(1 - e^{-w_w}\right)^\theta + \left(1 - e^{-w_h}\right)^\theta, \tag{6}$$

where $w_w$, $w_h$, $\rho_x$, $\rho_y$ is calculated as shown below:

$$w_w = \frac{|w - w^{gt}|}{\max(w, w^{gt})}, \tag{7}$$

$$w_h = \frac{|h - h^{gt}|}{\max(h, h^{gt})}, \tag{8}$$

$$\rho_x = \left(\frac{b_{c_x}^{gt} - b_{c_x}}{c_w}\right)^2, \tag{9}$$

$$\rho_y = \left( \frac{b_{c_y}^{gt} - b_{c_y}}{c_h} \right)^2, \tag{10}$$

The SIoU function is a non-squared loss function that helps the model find smaller deviations between predicted and actual values, improving the performance of the citrus fruit detection model with faster convergence. Therefore, SIoU is used as a new loss function instead of CIoU in this paper.

### 3.2.3. Context Transformer Self-Attention Module

In order to further improve the accuracy of citrus fruit recognition and localization while keeping the model lightweight, this paper performs the fusion of the CoT (Contextual Transformer) module for the backbone network of the YOLOv5s model.

The backbone feature extraction network of the YOLO series is the CNN network, which has a translation invariance and localization, and lacks the ability of global long distance modeling. In response, Tao Mei et al. creatively integrated the dynamic contextual information aggregation of the self-attention mechanism of the Transformer framework in the field of natural language processing with the static contextual information aggregation of convolution [19], and proposed a plug-and-play CoT (Contextual Transformer) module. Using this module in combination with CNN networks, thus implementing a Transformer-style backbone called CoTNet (Contextual Transformer Networks), leads to a significant performance improvement with a constant number of parameters.

The structure of the modules before and after the improvement is shown in Figures 5 and 6. In the traditional self-attention module, the Key, Query and Value are obtained by a 1 × 1 convolution, and the multiplication of the pairs of Key and Query ignores the context information in the neighboring Key. In contrast, in the CoT module, the Key and Query take the original value directly, and the adjacent Key undergoes a k × k convolution to obtain the context information and connects it with the Query module. Then, the connected result is convolved by two 1 × 1 convolutions to obtain the attention matrix, and finally multiplied with the value after the 1 × 1 convolution to output the result. This result fully exploits the contextual information between neighboring Keys compared with the result obtained by the self-attention module, thus enhancing the visual representation.

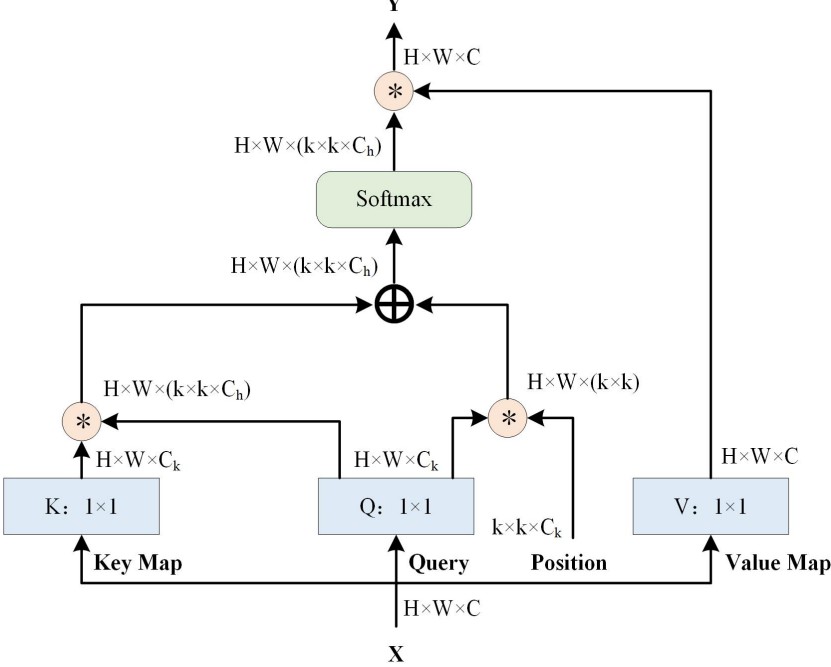

**Figure 5.** Traditional self-attention module.

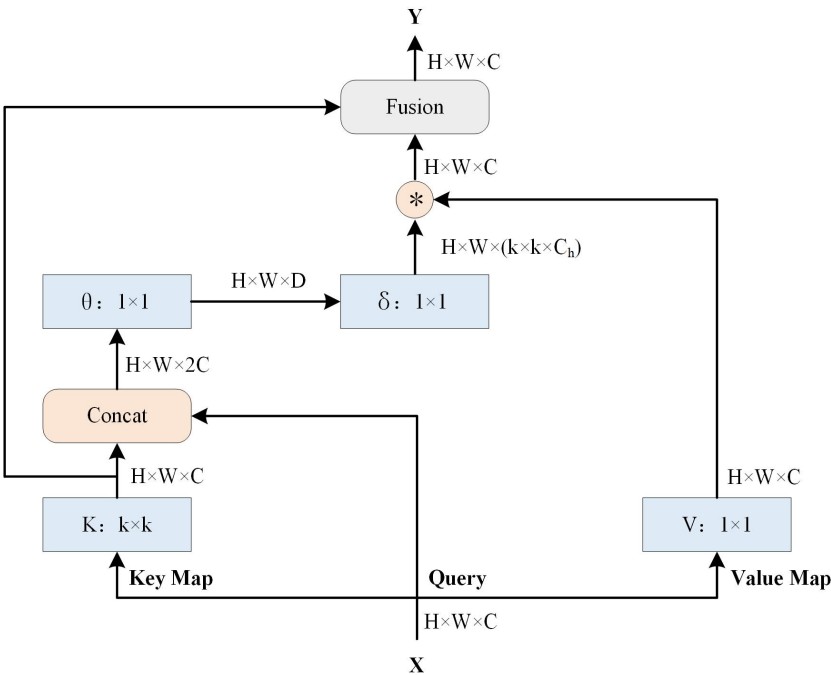

**Figure 6.** Contextual Transformer (CoT) module.

In this paper, we use CoT modules to replace all C3 modules in the YOLOv5s backbone network to take full advantage of the rich context between adjacent keys and improve the accuracy of the model for the citrus fruit detection while maintaining a light weight.

## 4. DeepSort Algorithm

### 4.1. Principle of DeepSort Algorithm

The core of the traditional Sort algorithm is the Kalman filtering and the Hungarian algorithm. The role of the Kalman filter is to predict the new position of the target frame in the next frame, and the role of the Hungarian algorithm is to match the predicted frame obtained from the Kalman filter with the actual target frame obtained from the target detection for IOU, and assign an ID number to each successfully matched target to achieve the effect of tracking. However, this algorithm is prone to the problem of ID switching when overlapping or occlusion occurs between targets in the process of matching detection and prediction frames, resulting in poor counting results.

To address these problems, the DeepSort algorithm adds Matching Cascade to the Sort algorithm. The cost matrix [9] is obtained by calculating the Mahalanobis distance $d^{(1)}$ and cosine distance $d^{(2)}$ between the predicted target and the detected target, as shown in Equations (11) and (12), and then the detected result is matched; if the match is unsuccessful, the target is considered as a new target and a new ID number is assigned to the target. Otherwise, the ID number of the target remains unchanged. The DeepSort algorithm reduces the problem of frequent ID switching in the case of multiple occlusions. The DeepSort algorithm reduces the problem of frequent ID switching and improves the accuracy of multi-target tracking counting.

$$d^{(1)}(i,j) = (d_j - y_i)^T S_i^{-1} (d_j - y_i),\tag{11}$$

where $d_j$ denotes the information of the $j$th detection frame, $y_i$ denotes the target prediction frame position of the $i$th path, and $S_i$ denotes the covariance matrix.

$$d^{(2)}(i,j) = \min\left\{1 - r_j^T r_k^{(i)} | r_k^{(i)} \epsilon R_i\right\},\tag{12}$$

where $r_j$ denotes the feature vector of each detection target $d_j$ and $r_k$ represents the set of the last 100 frames of feature vectors.

### 4.2. DeepSort Algorithm Combined with YOLOv5 Algorithm

The algorithm flow of DeepSort combined with YOLOv5s is shown in Figure 7, and the specific steps are:

(1) Train the citrus fruit detection model using the improved YOLOv5s algorithm.

(2) Input the video frames into the trained model for detection and input the obtained borders and features into DeepSort.

(3) DeepSort uses Kalman filtering to obtain the predicted frames. It obtains a high accuracy of matching results by cascade matching and IOU matching.

(4) Calculate the Mahalanobis distance and cosine distance between the prediction frame and the detection frame to derive the cost matrix and perform the Hungarian algorithm matching.

(5) Output the results and perform the parameter update using Kalman filtering.

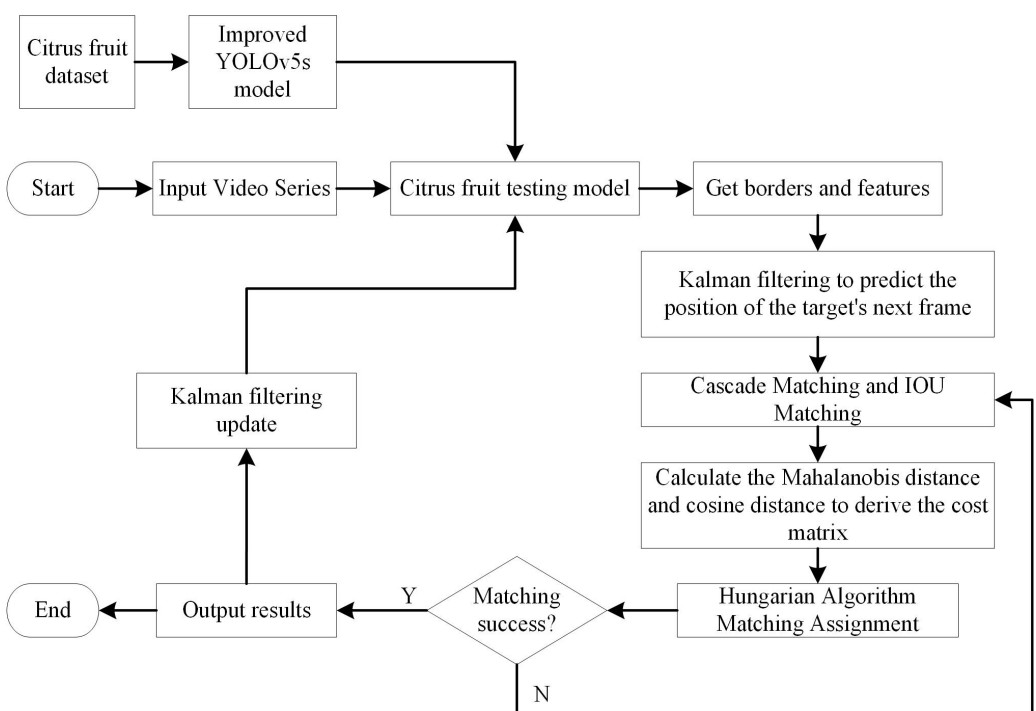

**Figure 7.** Algorithm flow chart.

## 5. Test and Analysis

### 5.1. Test Environment and PARAMETER Settings

This experiment is based on a Windows 10 operating system. In terms of the hardware configuration, the CPU is Intel i7-10875H and the GPU is NVIDIA GeForce RTX 2060. In terms of the software configuration, the Python version is 3.7.0, the PyTorch version is 1.12.0 and the CUDA version is 11.3. The image input during training The size is 640 × 640, the learning rate is set to 0.01, the number of samples in a single batch is set to 16, the momentum factor is 0.9, and the total number of training rounds is 300.

### 5.2. Evaluation Indicators

For target detection, Precision P, Recall R, FPS (Frames Per Second) and average recognition accuracy AP (Average Precision) are selected as the evaluation indexes of the detection effect of the YOLOv5s model.

The formulas of Precision P, Recall R, FPS (Frames Per Second) and average recognition accuracy AP (Average Precision) are expressed as:

$$\text{Precision} = \frac{T_p}{T_p + F_p}, \tag{13}$$

$$\text{Recall} = \frac{T_p}{T_p + F_N}, \tag{14}$$

$$\text{AP} = \int_0^1 \text{Precision} \cdot \text{dRecall}, \tag{15}$$

$$\text{FPS} = \frac{\text{TotalTime}}{\text{NumFigure}}, \tag{16}$$

where $T_p$ refers to the number of correctly detected samples, $F_p$ refers to the number of incorrectly detected samples, recall R is the ratio of the number of correctly detected samples to the number of samples in the dataset and FPS indicates the network detection speed.

For multi-target tracking, Multiple Object Tracking Accuracy (MOTA) is selected as the evaluation index of DeepSort, which reflects the false detection, missed detection and ID switching during tracking.

The Multi-Objective Tracking Accuracy (MOTA) is expressed by the formula:

$$\text{MOTA} = \frac{\sum_{i=1}^n \left( T_{P_i} - F_{P_i} - IDSW_i \right)}{\sum_{i=1}^n \left( T_{P_i} + F_{N_i} \right)}, \tag{17}$$

*5.3. Ablation Test*

To test the effectiveness of the improvement methods in this paper, ablation tests were conducted for the three improvement methods proposed in this paper, as shown in Table 1. The single improvement, two-by-two combination improvement and overall improvement were carried out on the YOLOv5s model, and the improvement effects were compared.

**Table 1.** Ablation test.

| Algorithm | CBAM | SIoU | CotNet | Precision | Recall | Average Precision | FPS |
|---|---|---|---|---|---|---|---|
| YOLOv5s | × | × | × | 87.42% | 86.04% | 91.75% | 86 |
| YOLOv5s+CBAM | ✓ | × | × | 90.58% | 83.75% | 92.66% | 82 |
| YOLOv5s+SIoU | × | ✓ | × | 89.07% | 87.87% | 92.99% | 90 |
| YOLOv5s+CotNet | × | × | ✓ | 88.47% | 86.01% | 93.28% | 85 |
| YOLOv5s+CBAM+SIoU | ✓ | ✓ | × | 91.11% | 89.13% | 94.11% | 87 |
| YOLOv5s+CBAM+CotNet | ✓ | × | ✓ | 87.03% | 86.96% | 92.50% | 80 |
| YOLOv5s+SIoU+CotNet | × | ✓ | ✓ | 90.84% | 89.70% | 94.77% | 89 |
| YOLOv5s+CBAM+SIoU+CotNet | ✓ | ✓ | ✓ | 91.21% | 90.25% | 95.26% | 84 |

As observed in Table 1, in a single improvement, the average accuracy of all three improvement methods has improved significantly, with a maximum improvement of 1.53%. In terms of the real-time performance, the addition of the CBAM module and the fusion of the CotNet module make the model detection speed decrease, and the real-time performance of the algorithm decreases. Moreover, the replacement of the SIoU loss function makes the real-time performance of the algorithm improve. In the two-by-two combination improvement, replacing the SIoU loss function and fusing the CotNet module makes the average accuracy improve further, and the mAP increases to 94.77%. Adding the CBAM module and replacing the SIoU loss function also provides better recognition results compared to the single improvement. However, with the combination of adding the CBAM module and incorporating the CotNet module, the average accuracy is reduced by 0.16% compared to

the single improvement with the smallest improvement. Finally, combining the above three improvements together, the mAP of the model detection reaches 95.26%, compared to 91.75% of the original model, with an average accuracy increase of 3.51%. The improved model detection speed is reduced by 2FPS compared to the pre-improvement, but still has a good real-time performance. It can be observed that the improved YOLOv5 model is better for citrus fruit identification and localization while maintaining a real-time performance.

### 5.4. Comparative Tests

In order to further verify the effectiveness of the improvement, the improved model is compared with the mainstream target detection algorithms at this stage in this paper for performance metrics. Under the same software and hardware environment, the same experimental parameters were set and the self-built citrus fruit dataset was used. The performance comparison results of each algorithm are shown in Table 2.

**Table 2.** Performance comparison of each algorithm.

| Algorithm | Precision | Recall | Average Precision | Models/MB | FPS |
|---|---|---|---|---|---|
| YOLOv3-SPP | 90.13% | 90.52% | 87.25% | 126.1 | 80 |
| YOLOv3-Tiny | 86.35% | 88.84% | 80.83% | 18.3 | 104 |
| YOLOv4-Tiny | 85.76% | 88.29% | 90.32% | 21.6 | 110 |
| YOLOv5s | 87.42% | 86.04% | 91.75% | 13.7 | 86 |
| Improved YOLOv5 | 91.21% | 90.25% | 95.26% | 17.8 | 84 |

As can be observed from Table 2, the highest precision, recall and average precision of the mainstream target detection algorithms at this stage are 90.13%, 90.52% and 91.75%, respectively. The improved model in this paper has improved 1.08 percentage points and 3.51 percentage points in accuracy and average precision, respectively, and the recall rate has also improved compared with that before the improvement. In terms of the model size, the improved model is slightly larger, but it is still a lightweight model. In terms of the real-time performance, the improved model decreases in network detection speed, but still maintains a fast speed, which can guarantee the real-time performance of the algorithm. Taken together, the improved algorithm improves the accuracy of model recognition while ensuring the lightweight effect and real-time performance, and shows a superior performance in the citrus fruit detection in complex environments.

### 5.5. Analysis of Results

In order to more intuitively reflect the performance of the improved algorithm, the citrus fruits were detected using the YOLOv5s algorithm before and after the improvement, as shown in Figure 8. Two sets of photos were selected: daytime with occlusion and nighttime with occlusion. From the detection results in both sets of photos, it can be observed that the improved YOLOv5s algorithm can identify more targets in both daytime and nighttime, and can identify targets that are occluded by branches and leaves without false detection, which is a better performance. This demonstrates that compared with the original algorithm, the improved algorithm has a higher recognition accuracy for citrus fruits and can achieve more accurate recognition and localization in the case of occlusion. Its network detection speed FPS is 84, which is reduced but still maintains a good real-time performance, indicating that the improved algorithm is more suitable for the recognition of citrus fruits in orchards.

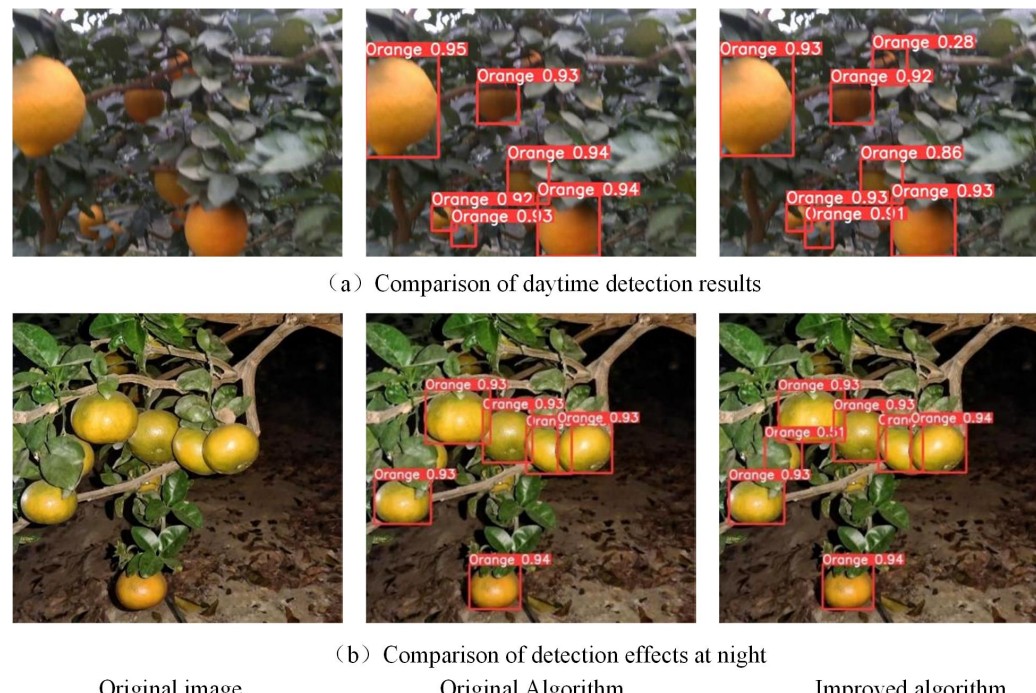

（a）Comparison of daytime detection results

（b）Comparison of detection effects at night

|  Original image | Original Algorithm | Improved algorithm |

**Figure 8.** Comparison of the detection effect of the model before and after improvement.

Meanwhile, the information of citrus fruits detected by the improved YOLOv5s algorithm is input into the DeepSort algorithm, and the counting of citrus fruits can be realized, as shown in Figure 9.

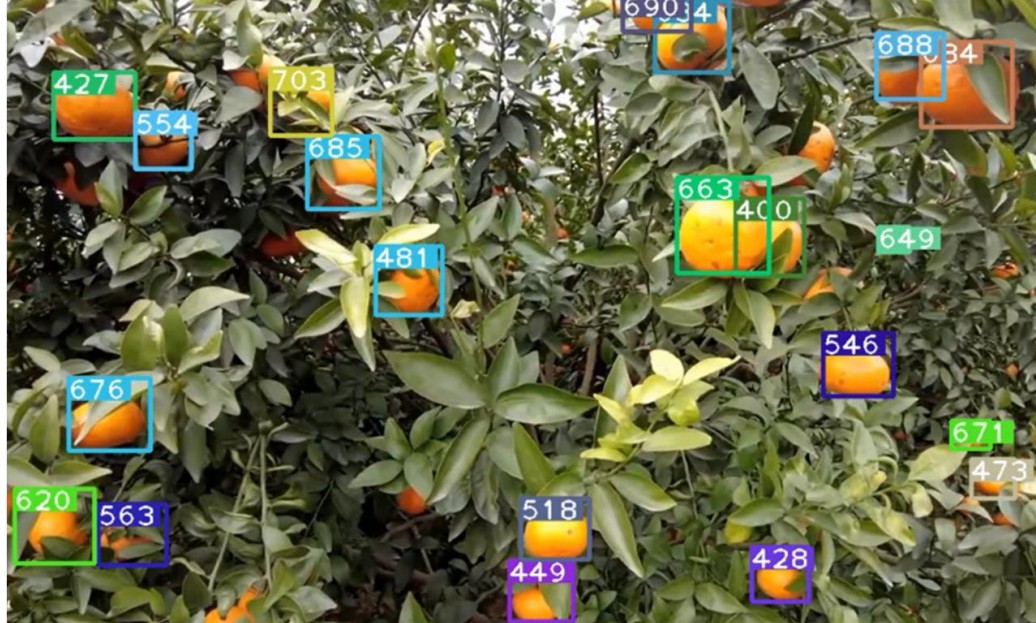

**Figure 9.** Counting effect.

In order to verify the impact of the YOLOv5s algorithm improvement on the performance of the DeepSort algorithm before and after, ten segments of citrus fruit videos were selected and input into the algorithm, and their performance metrics, Multi-Objective Tracking Accuracy (MOTA), were compared in this paper; the results are shown in Table 3.

**Table 3.** Comparison of DeepSort performance metrics.

| Video Serial Number | MOTA before Improvement | Improved MOTA |
|---|---|---|
| 1 | 87.37% | 93.30% |
| 2 | 89.62% | 89.25% |
| 3 | 89.05% | 92.63% |
| 4 | 86.26% | 89.15% |
| 5 | 90.83% | 91.03% |
| 6 | 89.73% | 89.91% |
| 7 | 87.18% | 89.18% |
| 8 | 91.31% | 90.69% |
| 9 | 89.34% | 93.34% |
| 10 | 88.76% | 89.85% |
| Average MOTA | 88.95% | 90.83% |

Comparing the evaluation index MOTA before and after the improvement, the average multiple object tracking accuracy increased from 88.95% to 90.83%, which is an improvement of 0.88 percentage points, indicating that the improvement of the detection algorithm not only enhanced the recognition and localization effect of citrus fruits, but also led to the improvement of the accuracy of fruit counting.

Meanwhile, in order to verify the impact of the improved method in this paper on the real-time performance of the overall system, the overall model before and after the improvement was compared in terms of the number of frames per second detected, and the results are shown in Table 4.

**Table 4.** Comparison of overall model detection frames per second.

| Models | Frames per Second |
|---|---|
| YOLOv5s + DeepSort | 78 |
| Improved YOLOv5s + DeepSort | 72 |

The overall model execution speed dropped slightly after the improvement, with FPS dropping from 78 to 72, but still managed to maintain a good real-time performance.

**6. Summary and Outlook**

In this paper, we propose an improved YOLOv5s algorithm and combine it with the DeepSort algorithm to achieve the recognition and counting of citrus fruits by adding a CBAM attention mechanism, replacing the SIoU loss function and integrating the CotNet module in the backbone network. This is in order for the model to achieve better results in the recognition and localization of citrus fruits while maintaining a better real-time performance. The detection results obtained after the improvements are fed into DeepSort, which makes the counting effect improve as well. The experimental results demonstrate that the improved YOLOv5s algorithm achieves an average accuracy of 95.26% in the self-built citrus dataset, which is 3.51% higher than the original algorithm, and the counting accuracy of the DeepSort algorithm is also 0.88% higher, with a decrease in the network detection speed.

This paper proposes a citrus fruit recognition and counting algorithm that improves YOLOv5s and combines DeepSort to achieve a real-time detection and tracking of citrus fruits in complex environments, improving the recognition and counting of citrus fruits while maintaining the light weight of the model and the real time of the algorithm. This helps assist staff in managing orchards, reduces labor costs, and achieves an intelligent fruit yield estimation in natural environments.

Synthesizing the current research status in the field of fruit yield estimation and target detection, the following outlook is provided for future research:

(1) Further detection of fruit ripeness. Although this paper achieves fruit counting, it does not distinguish fruit maturity, and there is still much room for improvement in the accurate estimation of the actual fruit yield.

(2) Improve the detection speed of the model to ensure the real-time performance of the algorithm. The improved algorithm in this paper has a decrease in real time, and the accuracy of the target detection cannot be pursued only. The performance of each aspect should be balanced.

(3) Further optimization of small target detection. The algorithm used in this paper does not take into account the small target detection aspect and needs to be further optimized.

**Author Contributions:** Conceptualization, J.X.; methodology, Y.L.; software, W.H.; validation, Y.L.; formal analysis, Z.Z.; investigation, Y.L.; resources, Z.Z.; data curation, Y.L.; writing—original draft preparation, Y.L. and W.H.; writing—review and editing, J.X. and W.H.; visualization, Y.L.; supervision, J.X.; project administration, J.X.; funding acquisition, South China Agricultural University. All authors have read and agreed to the published version of the manuscript.

**Funding:** This work is funded by the National Natural Science Foundation of China (Project No. 32071912), open competition program of the top ten critical priorities of the Agricultural Science and Technology Innovation for the 14th Five-Year Plan of Guangdong Province (2022SDZG03), and Special Funds for the Cultivation of Guangdong College Students' Scientific and Technological Innovation. ("Climbing Program" Special Funds) (pdjh2022b0079).

**Data Availability Statement:** Data are contained within the article.

**Acknowledgments:** The authors wish to thank the anonymous reviewers for the useful comments in this paper.

**Conflicts of Interest:** The authors declare no conflict of interest.

## Abbreviations

The following abbreviations are used in this manuscript:

| | |
|---|---|
| YOLO | You only look once |
| CBAM | Cost Benefit Analysis Method |
| IOU | Intersection over Union |
| SORT | Simple Online Realtime Tracking |
| Cot | Contextual Transformer |
| CotNet | Contextual Transformer Networks |

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
