# Peer review of "Citrus Identification and Counting Algorithm Based on Improved YOLOv5s and DeepSort"

_agronomy, doi:10.3390/agronomy13071674_

Round 1

Reviewer 1 Report

In the manuscript entitled “Citrus identification and counting algorithm based on improved YOLOv5s and DeepSort”, the authors propose improving the YOLOv5s network model commonly used for counting citrus in complex environments. The improved model incorporates the CBAM attention mechanism module, the Contextual Transformer self-attention module, and the SIoU loss function. The comparing results from the original and the improved model are discussed. The improved model showed an accuracy increase of 3.51%, but detection speed is reduced by 2FPS; additionally, the average multiple object tracking accuracy increased by 0.88 percentage points.

Some observation:

1)    The improved model's performance slightly outperformed the original; however, the authors do not mention the execution time taken for data processing of both models.

2)    Summary and Outlook: the authors should mention how the improvement of the model impacts the counting of citrus fruits.

3)    Some minor typos: line 67 (, validation set and set test).

Reviewer 2 Report

I have completed a thorough review of the manuscript. While I find the overall topic and approach of the study  interesting, I have several comments:

1-Authors should add one sentence to the abstract about the limitations of the method or future research gap that they mention in the conclusion.

2- The introduction provides sufficient background information on the challenges of citrus fruit counting in orchards and the existing methods in the field. However, it would be beneficial to extend it with home journal papers.

3- Authors briefly mention the validation using a self-collected dataset, but it lacks essential details therefore it needs more details.

4-  Overall language and clarity of the manuscript are good. however, authors need to define each abbreviation in the first use. 

Round 2

Reviewer 2 Report

Thank you for revising the paper and for your response.